# Exploring the Space of Possibilities in Cascading Disasters with Catastrophe Dynamics

**DOI:** 10.3390/ijerph17197317

**Published:** 2020-10-07

**Authors:** Arnaud Mignan, Ziqi Wang

**Affiliations:** 1Institute of Risk Analysis, Prediction and Management (Risks-X), Academy for Advanced Interdisciplinary Studies, Southern University of Science and Technology (SUSTech), Shenzhen 518055, China; 2Department of Earth and Space Sciences, Southern University of Science and Technology (SUSTech), Shenzhen 518055, China; 3Earthquake Engineering Research and Test Center, Guangzhou University, Guangzhou 510405, China; ziqidwang@yahoo.com

**Keywords:** catastrophe dynamics, absorbing Markov chain, topology, natural hazard, anthropogenic hazard, network failure, business interruption, disease, critical infrastructure, social unrest

## Abstract

Some of the most devastating natural events on Earth, such as earthquakes and tropical cyclones, are prone to trigger other natural events, critical infrastructure failures, and socioeconomic disruptions. Man-made disasters may have similar effects, although to a lesser degree. We investigate the space of possible interactions between 19 types of loss-generating events, first by encoding possible one-to-one interactions into an adjacency matrix A, and second by calculating the interaction matrix M of emergent chains-of-events. We first present the impact of 24 topologies of A on M to illustrate the non-trivial patterns of cascading processes, in terms of the space of possibilities covered and of interaction amplification by feedback loops. We then encode A from 29 historical cases of cascading disasters and compute the matching matrix M. We observe, subject to data incompleteness, emergent cascading behaviors in the technological and socioeconomic systems, across all possible triggers (natural or man-made); disease is also a systematic emergent phenomenon. We find interactions being mostly amplified via two events: network failure and business interruption, the two events with the highest in-degree and betweenness centralities. This analysis demonstrates how cascading disasters grow in and cross over natural, technological, and socioeconomic systems.

## 1. Introduction

As humans increase their impact on the planet, the risk associated with natural and man-made hazards can be amplified by emerging chains-of-events [1,2,3,4,5,6,7,8]. With rapid urbanization and lifeline connectivity growth, secondary, tertiary, quaternary, and further consequences of initial triggers are likely to occur more often, and more severely. Despite being among the highest-impact threats that society faces, the range of possible domino effects remains mostly unforeseen and unmodelled, due to the interacting process being highly complex, inter-systematic, and only partially experienced. This issue has been recognized recently as a barrier to proper multi-risk governance [9,10,11].

Some of the most devastating initial triggers include earthquakes and tropical cyclones, due to the tremendous energy they release and their wide spatial footprints [12]. Common secondary effects include other natural events, such as landslides and tsunamis [13], as well as critical infrastructure failures, such as industrial accidents and network breakdowns [14]. All those events and their consequences may further lead to socioeconomic disruptions, such as business interruption, social unrest, healthcare degradation, and economic crisis. The most recent infamous example of such consequences is the COVID-19 pandemic [15].

Although the ultimate goal of catastrophe risk assessment should be the probabilistic quantification of all physically imaginable interactions between the natural, built and cyber environments, and this at a global scale, the scope is so vast that a brick-by-brick approach should be favored at the present time. Directions being currently investigated include: (i) site-specific multi-risk engineering modelling at the local level [7,16]; (ii) detailed exploration of a given interaction system, e.g., a natural system [2] or a natural–technological interface (i.e., Natech) [6]; (iii) detailed exploration of the full chain-of-events cross-system for a given trigger, e.g., an earthquake [5,13,17]; and (iv) general rules of catastrophe dynamics based on theory and on simplified, generic perils and interactions [4,18,19]. We will here follow the latter approach. In particular, we are interested in exploring the space of possible cascading effects by learning from concepts of system dynamics, interaction-graph topology, and empirical analysis.

## 2. Materials and Methods

### 2.1. Catastrophe Dynamics

We employ the term “catastrophe dynamics” as it is used in [18], which describes an approach to understand and quantify all direct and indirect interactions between events during the temporal development of a catastrophe via system dynamics. It is only loosely related to “catastrophe theory”, which studies specific potential functions [20]. With catastrophes here defined as the accumulation of individual, loss-generating events that interact with each other, we assume that the dynamic process is described by the following linear differential Equation:(1)$$\frac{d\vec{X}\left(t\right)}{dt}=Q\vec{X}\left(t\right)$$

In which the dynamics of the vector $\vec{X}$ of events is fully described by the transition rate matrix Q that encodes all possible one-to-one direct/infinitesimal interactions. The propagator for $\vec{X}\left(t\right)\to \vec{X}\left(t+\tau \right)$ can be written as P(τ)=exp(Qτ), with the entries $p_{ij}\left(\tau \right)$ interpreted as the conditional probability of event j being triggered by event i for a period of τ. This study presumes that we have a direct estimate on the adjacency matrix, defined as A:=P(τ=1)=exp(Q), i.e., the matrix exponential of Q [21]. This presumption indicates that the probabilities for events i→j within a unit amount of time can be estimated, a construction that does not influence the generality of this study. It also assumes that Q is constant (i.e., stationary one-to-one interactions) and that probabilities are conserved with $\overset{n}{\underset{j=1}{\sum }}p_{ij}\left(\tau \right)=1$, or equivalently, $\overset{n}{\underset{j=1}{\sum }}q_{ij}=0$, where $q_{ij}$ denotes entries of the rate matrix Q.

This study aims at investigating emergent cascading patterns as the catastrophe’s dynamics evolve, given the specification of a $\left(n_{perils}+1)\times (n_{perils}+1\right)$ adjacency matrix A. As the individual one-to-one interactions between the events generated by $n_{perils}$ perils are a priori independent, we define an outflow event as a last matrix entry for conservation of probabilities with conditional probability $1-\overset{n_{perils}}{\underset{j=1}{\sum }}p_{ij}$. We also fix the last row of A, $p_{n_{perils}+1}=\left(0,\ldots ,0,1\right)$, i.e., the outflow event does not trigger any real event but itself. This means that a catastrophe will naturally die off, with the outflow event representing an absorbing state in Markov chain jargon. For convenience, we also define the reduced $n_{perils}\times n_{perils}$ adjacency matrix $\bar{A}$ without the outflow event (absorbing state).

The event interactions, which are encoded in A**,** can be represented by a graph G with a vertex set V of events and an edge set E of one-to-one interactions. Figure 1 shows an example of an interaction network encoded in A and displayed as G, in the case of a hydro-dam catastrophe system that includes both natural and technological hazards [4,22]. Each event can lead to dam failure along the chain of interactions. In the present study, we will use a wider event “grain” with all the elements of a hydro-dam concatenated within a so-called critical infrastructure failure (see Section 2.2). We will also solely consider loss-generating, triggering events and no event that could inhibit another event (naturally of via a mitigation measure).

Graphs can be described by various centrality measures, the most important ones being in-degree, out-degree, closeness, and betweenness [23]. Degree centrality of a vertex is the number of edges incident on the node; closeness centrality of a vertex is the average of the shortest path lengths from the vertex to all other vertices in the network; betweenness centrality of a vertex is the number of the shortest paths that pass through that vertex. In catastrophe dynamics, the main triggering events (or sources) are represented by high out-degree and closeness centralities; the main triggered events (or sinks) by a high in-degree centrality, and the main catalysts that promote cascading by a high betweenness centrality [16].

Applications in which catastrophic interactions are encoded in an adjacency matrix are as varied as in disease spread forecasting and risk mitigation [18,24], tramway infrastructure risk assessment [25], hydro-dam failure analysis [22], volcanic eruption post-crisis assessment [7], modeling of road network disruption by floods [26], or ecological disaster modeling [27]. The system’s dynamics is usually simulated, although an analytical solution to the final state can also be estimated [21]. While the analysis of the matrix power $A^{\tau }$ (or ${\bar{A}}^{\tau }$) for τ→∞ (i.e., discrete case, see [18] for the continuous case) suffices to explore the full space of possible interactions in a topological sense, we are here also interested in evaluating how a catastrophe can be amplified [19] in terms of chain-of-events length. This can be computed via the concept of fundamental matrix [21], which captures the frequency/intensity of each transition event before resting at the absorbing state.

With G finite, we can define the fundamental matrix
(2)$$N≔{\left(I-A\right)}^{-1}=I+\sum_{\tau =1}^{\infty }A^{\tau }=I+A+A^{2}+\cdots =I+M$$

With the entry $n_{ij}$, the expected number of times the chain-of-events reaches event j given that the catastrophe starts with event i and I the identity matrix. All events here represent transient states prior to the absorbing state being entered [21]. We then introduce the “interaction matrix” M, which excludes the step τ=0 that is independent of the topology of interactions. If A encodes 1→2 and 2→3, M additionally describes the chain 1→2→3. As τ increases, non-trivial patterns may emerge in M depending on the topology of A. This will be investigated in Section 3.1. Note that the reduced $n_{perils}\times n_{perils}$ interaction matrix $\bar{M}$, without the outflow event (absorbing state), will be displayed alongside $\bar{A}$ in the next two figures.

### 2.2. Historical Data Encoding

A wide variety of perils exist, which follow different processes and scales. Various taxonomies have been proposed [2,7] and we here adapt and extend the one described in [4]. The list of perils, 19 in total with short descriptors and identifiers, is given in Table 1. Natural events have a geological, geomorphological, hydrological, meteorological, biophysical, ecological, or extraterrestrial origin; anthropogenic events have a technological, economical, or social origin (no distinction is made between accidental and malicious).

Any taxonomy, by construction, is subjective. We attempted to limit possible discrepancies by keeping event classes as generic as possible, discriminated by different physical processes, and as close as possible as proposed classes in the literature [2,4,7]. We also narrowed the study to events which have an impact from the scale of a city to that of a continent, are relatively sudden and not excessively rare. Hence, we did not consider common or freak accidents (road, domestic, workplace events), long-term trends related to climate change [28] and ecological disasters [27], or speculative and existential risks [29]. As the focus of our work is on hazard and risk interactions, we also defined events so that there can be a clear, explicit one-to-one interaction between two events. All events, direct or indirect, also lead to direct losses defined in terms of casualties and/or economic losses.

Historical cases of catastrophes amplified by cascading effect abound. Table 2 lists some of the most infamous but also other, less known cases that provide some evidence for other potential interactions. Our aim is to explore the space of possibilities for cascading disasters on Earth, first by determining which cells of A are non-zero and second by calculating M. Note that quantifying such interactions in terms of explicit conditional probabilities $p_{ij}$ would require large databases, which are usually only limited to secondary effects, e.g., landslides and tsunamis triggered by earthquakes [5,13]. We follow a binary approach instead, with $p_{ij}$ either 0 or non-zero (see Section 4.1 on a discussion of such a limitation). The literature sources that led to Table 2 are given in Appendix A.

## 3. Results

We apply the rules of catastrophe dynamics to better understand the range of possible cascading effects. We first explore how different one-to-one interaction topologies yield emergent cascading behaviors based on Equation (2) (Section 3.1). Once understood at a general level, we investigate the topology of one-to-one interactions derived from historical data (Table 2) to identify in a semi-quantitative way how the space of possibilities expands in realistic conditions and in which cases feedback loops leading to risk amplification can be expected (Section 3.2).

### 3.1. Topological Characteristics of Cascading Effects

We test 24 different topologies of one-to-one interactions in a 10×10 square matrix $\bar{A}$ with constant conditional probability $p_{ij}=p$ fixed for all non-zero entries (the matching matrix A with outflow event for conservation of probabilities is thus of dimensions 11×11). The selected topologies, shown in Figure 2, represent a wide range of possible configurations with different (no-trivial) emergent properties in the interaction matrix M (or $\bar{M}$ on the figure). The topological configurations were defined with the following logic: the patterns increase in complexity per descending row in Figure 2, starting with events triggering themselves (diagonal ${\bar{A}}_{ii}$, 1-to-1 interactions), followed by cases of only one trigger or of only one triggered event (1-to- n or n-to-1 interactions), and then by cases of two triggers or two triggered events (2-to-n or n-to-2 interactions). Starting with row 3 of Figure 2, any event can be a trigger and/or be triggered, with the number of possibilities increasing on average in subsequent plots (n-to-n interactions). We also included gaps in the patterns of each configuration in order to investigate the role of specific indices of A. No automated rule was applied with the role of Figure 2 solely to illustrate how M can derive from A geometrically. We used max τ=10 at which time step most emergent cascades (i.e., matrix M) have stabilized (although cases 1–20 were already stable at τ=3). The graphs G of the various matrices $\bar{A}$ can be found in Appendix B for different centrality measures (Appendix B: Figure A1, Figure A2, Figure A3 and Figure A4). We observe two cascading phenomena, an increase of the space of possibilities and an amplification of some interactions. Although these results simply derive from the matrix power and summation operators Equation (2), some of the emerging patterns are non-trivial:

(i) The feasible set of interactions expands in the subspace (i,j) (mostly represented in blue in Figure 2) as soon as event i triggers a chain of at least one event (excluding i and j), which finally reaches event j. This yields to all possible permutations of (i,j) being explored, i.e., the cardinality of the set of cascades increases. It follows that emerging cascades (see cases 7–8, 12–13) remain rare when the number of triggers and/or triggered events is low (cases 1–13). As the numbers of potential triggers and triggered events increase, more transitions become feasible (cases 14–15, 20, 22–23). For random configurations of one-to-one interactions (cases 21a–b), the feasible set of transitions can rapidly expand by the merging of several subspaces being explored. In short, as physical couplings increase, cascades become richer, and multi-risk assessment more prone to negative surprises if the underlying models fail to correctly encode the interactions related to intermediary events between i and j. The impact on overall risk may remain limited however due to the exponential decay of triggering probabilities at each step of the stochastic process ($p_{ij}<p$ represented in blue in Figure 2).

(ii) Amplification effects (represented in warm colors in Figure 2) occur as soon as an event can trigger itself (case 1). Such feedback loop $p_{ii}>0$ further amplifies, via propagation, the other events that it can trigger (compare cases 3–5 to non-amplification cases 2–4, the same rule being observed for more complex topologies, such as cases 6–15). When there is no such feedback loop, amplification also occurs if event j is triggered by an event i that triggers an intermediary event, which also triggers j, the best examples being cases 17 and 19. Finally, both possibilities can combine their effec ts for further amplifications (cases 16 and 18, extreme cases 22–24). The less trivial behaviors are observed in cases 14–15 and 20–21. For multi-risk assessment, the amplification can be understood as an increase in the re-occurrence of event j in a catastrophic chain of events starting with event i. As a corollary, losses associated to the said event accumulate as well, leading to catastrophic loss amplification [19].

This semi-quantitative analysis illustrated how both the space of possible cascading effects and overall loss can increase via a small number of linear algebra rules, albeit in a non-trivial way. We will now investigate which is the overall configuration of A in reality and which its expected impact is on cascading disasters on Earth.

### 3.2. Application to Cascading Disasters Based on Historical Data

We now encode the one-to-one interactions of Table 2 into an adjacency matrix A and compute the matching interaction matrix M. As in Section 3.1, the interactions, here listed in the last column of Table 2, are described in a binary approach, $p_{ij}$ fixed to an ad-hoc p if observed and to zero otherwise. Note that we made the choice to encode interactions based on past observations for the advantage of being relatively objective. This approach could be complemented by including additional plausible interactions via expert judgement [7] and reasoned imagination [4]. Although the encoding is likely incomplete (see Section 4.1), it still represents a reasonable overview of possible interactions across the natural, technological, and socioeconomic systems, which should be further refined in future works. Results are shown in Figure 3.

We show that the space of possibilities expands, i.e., is enriched by additional transient events, for the technological and socioeconomic systems for any type of trigger. This is mainly controlled by network failures (NF) and business interruptions (BI) as they can be triggered by most possible events and in turn can trigger many events in the technological and socioeconomic systems. Diseases (DI) also emerge systematically, whichever the initial trigger. The potential of surprising chains-of-events leading to secondary natural perils is very limited in comparison. We also verify the independence of extraterrestrial events.

Amplification effects (i.e., increased number of times a cascade goes through event j given the initial event i) are most pronounced for i an earthquake (EQ) or a network failure (NF). In both cases, feedback loops can occur such as large aftershocks amplifying risk [17] and cascading network failures [30]. Moreover, both great earthquakes and lifelines have wide footprints, which can increase the chance of other feedback effects via various types of intermediary events (see examples in Table 2). For triggered events j, the main amplification effects are observed for network failures (NF) and business interruptions (BI) as they can be triggered by many different perils, which, via various paths, may lead to a repeat of those two events.

Figure 4 finally shows the graph G derived from the empirical adjacency matrix $\bar{A}$, illustrating the complexity and strong coupling of peril interactions. The highest in-degree and betweenness centralities are observed for network failure (NF) and business interruption (BI), which is in agreement with the results obtained above. Both events are important targets and catalysts of further interactions in the catastrophe network, explaining their critical role in cascading. In particular, they both provide an interface between loss-generating events in the natural system on one side and in the technological and socioeconomic systems on the other side.

## 4. Limitations and Future Directions

### 4.1. Incompleteness Issues in Historical Data

This study is based on the analysis of only 29 historical cases of cascading disasters (Table 2). Although believed to be representative of a relatively large space of physically possible interactions, other potential interactions, but also perils, are necessarily missing. Moreover, any given one-to-one interaction present in Table 2 recurs less than 29 times, none being systematic. This is the reason why we used a binary, semi-quantitative approach by fixing a constant ad-hoc conditional probability p, which led to an unweighted adjacency matrix. With all interactions considered equal, the ones occurring rarely take a preponderant role relative to other, more common, interactions. The best example is the case of a landslide triggering a disease outbreak (1994 Northridge earthquake case, one of the most peculiar chains-of-events listed in Table 2) which is extremely rare and only included here to show the extent of possible interactions. Although it is likely to lead to biases in event amplification in Figure 3, it does not further cascade into M, and hence does not impact on our main conclusions. The other interactions are unlikely to lead to any significant bias in relative amplification measures.

Encoding the adjacency matrix with realistic values of $p_{ij}$ will require extensive data mining in historical records to obtain reliable estimates from a more complete set of data. Such a database has yet to be built, making our approach a pilot study in the promotion of such development in the near future. A further level of incompleteness is related to potential intermediary interactions missing in the encoding of the raw data, as this process is subject to interpretability (the reason why we provide all the raw data in Appendix A). Here again, the development of a large database will help to identify the best event granularity and the general physical rules to consider in the definition of one-to-one interactions.

### 4.2. Beyond One-to-One Links

The full scope of catastrophe dynamics is not limited to the linear dynamic process represented by Equation (1), which provides only a first-order description. We observed in the literature important consequences that cannot be described by one-to-one links but by multiple variables (represented by ⇒ in Table 2). For instance, it is sometimes observed that an independent coinciding event amplifies risk, a textbook example being wind during or just after a non-atmospheric event: wind can amplify fires following an earthquake (firestorm [31]), wildfires (megafires [32]), or disease spread during a landslide (dust-borne spore clouds [33]). Rain can trigger a landslide where the soil has first been destabilized by a wildfire [8] or lahars following a volcanic eruption (e.g., tropical cyclone during the 1991 Mount Pinatubo eruption) [2].

Many one-to-one interactions could be further amplified by underlying conditions. It has, for example, been suggested that a civilization or society can collapse following a natural catastrophe, but only if the economic situation of the civilization or society is very weak [34]. Other long-term drivers such as climate change are also known to amplify those direct interactions by increasing the available energy by temperature rise [28]. Conflicts (e.g., a war), as observed in historical cases (Table 2), can be a direct consequence of another conflict (e.g., terrorism). However, as illustrated in Figure 3, no other chain of one-to-one interactions is likely to lead to conflict. Such an event is more likely to be triggered by the accumulation of many different, sometimes independent, events.

### 4.3. The Temporal Evolution of Cascading Phenomena

As temporal processes evolve, so do cascading phenomena: earthquakes could directly trigger fires prior to the 20th century when households were lighted by candles that could fall due to ground shaking (e.g., 1755 Lisbon earthquake [35]). Nowadays, direct fires are much less likely, occurring via critical infrastructure and network failures (e.g., gas leak). In other cases, much more interactions would be expected today: The 1859 geomagnetic storm (Carrington Event) did not lead to cascades at the time, but its reoccurring today would cause widespread electrical disruptions and damage, far more than the 1989 Québec event [36]; it is the same with the 1908 Tunguska airburst, which, if it had occurred over an urban area [37], would have led to critical chains-of-events as are observed following other wide-footprint high-energy events.

The limitations described just above and in Section 4.2 show the way to follow in this domain, i.e., adding non-linearity to Equation (1) (which could relate, then, more closely to catastrophe theory [20]), time-dependency in variables and conditional probabilities, as well as developing counterfactuals [38] in multi-risk. Analytical solutions such as Equation (2) would however become rarer with simulation approaches more adapted to model non-Markovian processes [17,19,24,39].

### 4.4. The Concept of Unit Time in the Current Framework

Finally, it is important to mention that the cascading events studied in this paper occur in a natural physical time scale, and the (physical) time required for each triggering event to happen varies. For example, the time required for an earthquake to trigger a tsunami can be different from that to trigger a critical network failure or a landslide. However, in the proposed discrete Markovian framework, the physical time scale is simplified into the “consecutive trials” in a Markov chain. In consequence of this simplification, only the causal relationship presented within the original physical time is preserved. For instance, the simplification suggests that both the triggering event “A → B” and “A → C” are counted as “a single trial”, with the causal relationship that “A” triggers “B” or “C” being preserved. To further incorporate the time difference into each event triggering, in future studies, one could attempt to recover the rate matrix in the continuous master equation—Equation (1)—without the presumption that adjacency matrix A was directly estimated.

## 5. Conclusions

We explored the range of potential cascading effects in catastrophes by applying the rules of system dynamics and by defining finite catastrophic chains-ofevents as absorbing Markov chains. We first investigated the richness of cascades and amplifying feedback effects by computing the interaction matrix M, i.e., a variant of the fundamental matrix N, for idealized topologies of one-to-one interactions encoded in an adjacency matrix A (Section 3.1). We found that cascade patterns rapidly become non-trivial (Figure 2), which is likely to hamper multi-risk assessment and mitigation when not all possible interactions have been considered. We then encoded 29 historical catastrophes known for their cascading behavior (Table 2, Appendix A) in A and computed the matrix M representative of a general, yet highly simplified, catastrophe dynamical system (Section 3.2). We observed a clear difference between the natural system and man-made systems (technological and socioeconomic) with cascades being enriched and amplified mostly via network failure and business interruption, which bridge the different systems (Figure 3 and Figure 4).

The present analysis should in the future be extended towards a systematic assessment of cascading effects for all types of perils and full chains of events. The key to such an approach would be the combined definition and categorization of both events and event-to-event interactions. The generic perils defined here (Table 1) shall be characterized in physical terms (e.g., energy type, scaling), and the physics of possible interactions made explicit. A physics-based ontology of peril interactions would provide the means to build a database with a proper attribute definition for quantitative multi-risk analysis.

## Figures and Tables

**Figure 1 ijerph-17-07317-f001:**
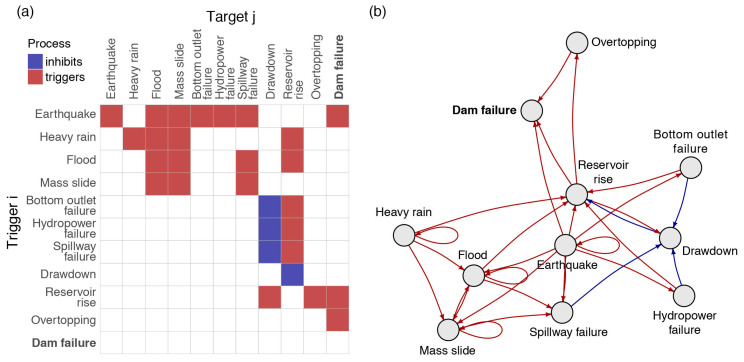
Illustration of how event interactions can lead to the catastrophic failure of a hydro-dam: (**a**) Interactions encoded in an adjacency matrix A following [4,22]; (**b**) Interactions described in a graph G.

**Figure 2 ijerph-17-07317-f002:**
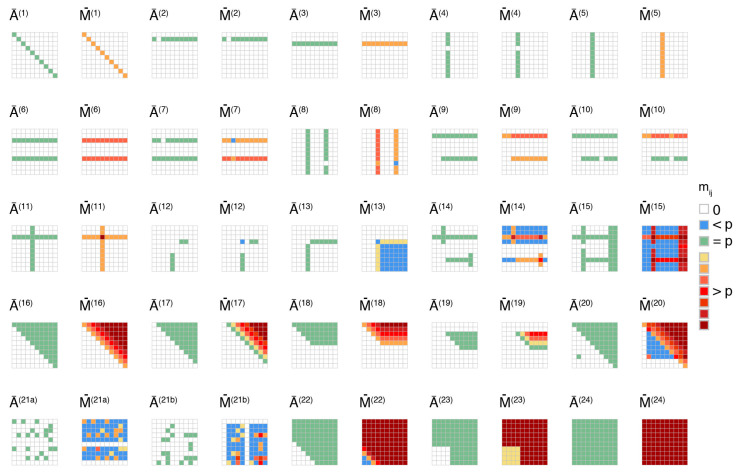
Different topologies l of one-to-one interactions in a reduced adjacency matrix ${\bar{A}}^{\left(l\right)}$ and matching reduced interaction matrix ${\bar{M}}^{\left(l\right)}\left(\max\;\tau =10\right)$. Warmer colors are proportional to the ad-hoc probability, here fixed to p=0.1.

**Figure 3 ijerph-17-07317-f003:**
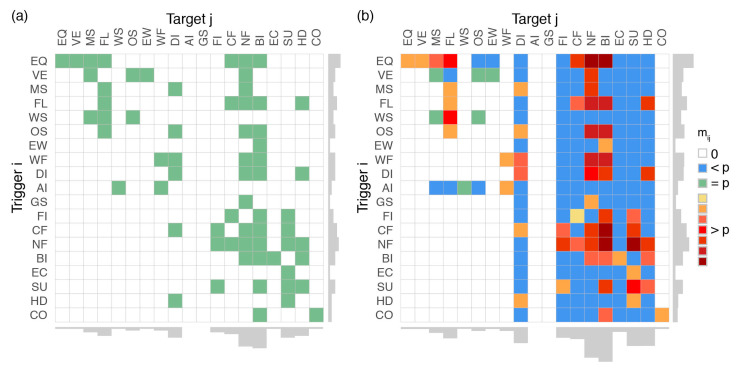
Topology of historical catastrophes constructed from Table 2. (**a**) Reduced adjacency matrix $\bar{A}$; (**b**) matching reduced interaction matrix $\bar{M}\left(\max\;\tau =3\right)$ showing both an expansion of the transient interaction set (mostly in blue) and an amplification of certain events for given initial triggers (warm colors). Warmer colors are proportional to the ad-hoc probability, here fixed to p=0.1.

**Figure 4 ijerph-17-07317-f004:**
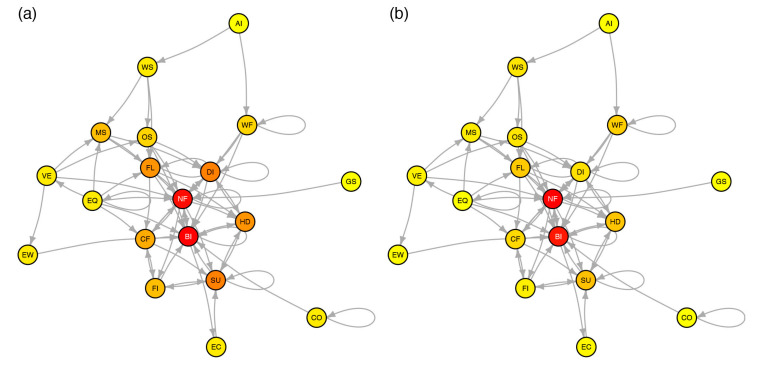
Catastrophe graph G of the reduced adjacency matrix $\bar{A}$ shown in Figure 3. (**a**) In-degree centrality; (**b**) betweenness centrality. A warmer color represents a higher centrality value; see Table 1 for peril identifier definition.

**Table 1 ijerph-17-07317-t001:** A generic taxonomy of perils.

ID	Peril	Including
NATURAL		
Geological and Geomorphological		
EQ	Earthquake	Shaking, fault rupture, ground displacement, subsidence
VE	Volcanic eruption	Explosive or effusive
MS	Mass slide	Landslide, rockfall, avalanche, mudslide, liquefaction
Hydrological		
FL	Flood	River flooding, tsunami, storm surge, flash flood
Meteorological		
WS	Windstorm	Tropical cyclone, extratropical windstorm, winter storm, tornado
OS	Other storm	Rainstorm, hailstorm, thunderstorm (lightning), ice storm, snowstorm, sandstorm, haze
EW	Extreme weather	Drought, heat wave, frost, extreme temperature gradient
Biophysical and Ecological		
WF	Wildfire	Bushfire, forest fire
DI	Disease	Outbreak, epidemic, pandemic
Extraterrestrial		
AI	Asteroid impact	Asteroid or comet, impact or air blast
GS	Geomagnetic storm	N/A
**ANTHROPOGENIC (Accidental or Malicious)**		
Technological		
FI	Fire	N/A
CF	Critical infrastructure failure	Explosion, toxic release (chemical, radioactive), water release (dam/levee break, overtopping)
NF	Critical network failure	In transportation, water and gas, electricity blackout, cyber-attack
Economical		
BI	Business interruption	In industry, agriculture, tourism, etc.
EC	Economic crisis	Recession, depression, financial crisis, hyperinflation
Social		
SU	Social unrest	Riot, strike, vandalism, looting
HD	Healthcare degradation	No rescue, security or access to hospital, unsanitary conditions, starvation (famine)
CO	Conflict	War, revolt, revolution, terrorism

**Table 2 ijerph-17-07317-t002:** List of infamous catastrophes with rich and/or peculiar chains-of-events.

Catastrophe ^1^	Observed Cascading Effects ^2^	Proposed Encoding ^3^
NATURAL TRIGGER		
Earthquake (EQ) as initial trigger		
2011 Tohoku, JP	EQ → great tsunami → nuclear disaster → blackouts, global nuclear energy turn-around ⇒ decline in global automobile production ⇒ significant fluctuations in global financial markets	(EQ, FL); (FL, CF); (CF, NF); (CF, BI)
2008 Wenchuan, CN	EQ → landslides → landslide lakes → downstream flooding EQ → energy networks outage → industrial production interrupted EQ → derailments → tanks collapse → fire EQ → landslides → roads damaged → rescues delayed EQ → chemical plant failure (toxic release) → health issues EQ → many businesses damaged	(EQ, MS); (MS, FL); (FL; FL); (EQ, NF); (NF, BI); (NF; CF); (CF, FI); (MS, NF); (NF, HD); (EQ, CF); (CF, DI); (EQ, BI)
2004 Sumatra, ID	EQ → tsunami → poor sanitation, lifelines ↓, tourism, fishing and farming ↓ EQ → great aftershocks EQ → volcanic eruption	(EQ, FL); (FL, NF); (FL, HD); (FL, BI); (EQ, EQ); (EQ, VE)
1994 Northridge, US	EQ → landslide → Valley Fever outbreak	(EQ, MS); (MS, DI)
1923 Kanto, JP	EQ → water network ↓ → fires → social unrest (Koreans attacked on false rumors) EQ → water network ↓ → unsanitary conditions → typhoid and dysentery ⇒ ideological, political and societal reforms ⇒ World War II	(EQ, NF); (NF, FI); (FI, SU); (NF, HD); (HD, DI)
1906 San Francisco, US	EQ → water and gas network ↓ → fires ⇒ 1907 Financial Panic	(EQ, NF); (NF, FI)
Volcanic eruption (VE) as initial trigger		
2010 Eyjafjallajokull, IS	VE → air travel disruption → airline bankruptcies	(VE, NF); (NF, BI)
2002 Stromboli, IT	VE → collapse of volcano side → tsunami → island closed to tourism	(VE, MS); (MS, FL); (FL, BI)
1783 Laki, IS	VE → extreme weather fluctuations → agriculture ↓ VE → poisonous cloud → sickness VE → poisonous cloud → agriculture ↓ (cattle mass death, scorched vegetation) → famine ⇒ climate change, 1789 French Revolution	(VE, EW); (EW, BI); (VE, OS); (OS, DI); (OS, BI); (BI, HD)
Mass slide (MS) as initial trigger		
1963 Vajont, IT	MS → tsunami on artificial lake → dam overtopping	(MS, FL); (FL; CF)
Flood (FL) as initial trigger		
2011 Thailand	FL → manufacturing sector ↓ → GDP ↓ FL → manufacturing sector ↓ → supply chain ↓ → international import sector ↓ → GDP ↓	(FL, BI); (BI, ES); (BI, NF); (NF, BI)
Windstorm (WS) as initial trigger		
2017 Hurricane Harvey, US	WS → rainfall event → fluvial inundation → transportation network ↓ → emergency response service ↓	(WS, OS); (OS, FL); (FL, NF); (NF, HD)
2012 Hurricane Sandy, US	WS → storm surge → electric network ↓ → health care facility evacuated WS → storm surge → power plants ↓ → electric network ↓ → other lifelines and supply chains ↓	(WS, FL); (FL, NF); (NF; HD); (FL, BI); (BI, NF); (NF, NF)
2005 Hurricane Katrina, US	WS → storm surge → levee failure → business interruptions WS → storm surge → levee failure → electrical cell phone stations ↓ → rescue, medical care, security scarce → violence and looting	(WS, FL); (FL; CF); (CF, BI); (CF, NF); (NF, HD); (HD, SU)
Other storm (OS) as initial trigger		
2008 southern China ice storm, CN	OS → roads, trains and flights ↓ → supply chains of energy and food ↓ → power plants shut down, food shortage ⇒ consumer price index ↑ nationwide	(OS, NF); (NF, NF); (NF, BI); (NF, HD)
1977 New York City thunderstorm, USA	OS → blackout → airports closed, rescue units and hospital operations hampered OS → blackout → looting and arson → fires → businesses damaged	(OS, NF); (NF, NF); (NF, HD); (NF; SU); (SU, FI); (FI, BI)
Extreme weather (EW) as initial trigger		
2006–2010 Syrian drought, SY	EW → crop failure → malnutrition → diseases EW → crop failure → food price ↑ → violence ⇒ 2011-present Syrian civil war	(EW, BI); (BI, HD); (HD, DI); (BI, ES); (ES, SU)
Wildfire (WF) as initial trigger		
2019–2020 bushfires, AU	WF → further bushfires WF → transportation and production ↓ → logistics, tourism ↓ WF → smoke-related diseases	(WF, WF); (WF, NF); (WF, BI); (NF, BI); (BI, BI); (WF, DI)
Disease (DI) as initial trigger		
2020 COVID-19	DI → healthcare system ↓ DI → travel restriction → tourism, energy, manufacturing sectors ↓ → GDP ↓, financial crisis DI → travel restriction → tourism ↓ → agriculture sector ↓ DI → travel restriction → domestic violence DI → secondary diseases	(DI, HD); (DI, NF); (NF, BI); (BI, ES); (BI, BI); (NF, SU); (DI, DI)
Asteroid impact (AI) as initial trigger		
1908 Tunguska, RU	AI → large fires ignited near ground zero AI → hurricane-force winds	(AI, WF); (AI, WS)
Geomagnetic storm (GS) as initial trigger		
1989 Québec, CA	GS → power blackouts → loss of sales → GDP loss	(GS, NF); (NF, BI); (BI, ES)
**ANTHROPOGENIC TRIGGER**		
Fire (FI) as initial trigger		
1986 Basel, CH	FI → chemicals released → water supply suspended	(FI, CF); (CF, NF)
Critical infrastructure failure (CF) as initial trigger		
1986 Chernobyl, UA	CF → radiation-related diseases, agricultural crops ↓ CF → nuclear fire	(CF, DI); (CF, BI); (CF, FI)
1984 Bhopal, IN	CF → gas-related diseases → workforce incapable of work, food supply shortages → food price ↑, transportation ↓ CF → vegetable crops affected, protests and violence	(CF, DI); (DI, BI); (BI, ES); (BI, NF); (CF, BI); (CF, SU)
1976 Seveso, IT	CF → chemical lesions, contaminated crops	(CF, DI); (CF, BI)
Network failure (NF) as initial trigger		
2003 blackout, IT	NF → Internet network ↓ → further power stations breakdown	(NF, NF); (NF, NF)
2003 blackout, US	NF → subway and water systems ↓, perishable food lost at restaurants and stores	(NF, NF); (NF, BI)
Social unrest (SU) as initial trigger		
1992 Los Angeles riot, US	SU → Fires → number of businesses lost SU → “mini-malls” destroyed, burglaries and vigilantes, health and fire service hampered	(SU, FI); (FI, BI); (SU, BI); (SU, SU); (SU, HD)
Conflict (CO) as initial trigger		
11 September 2001 terrorist attack, US	CO → business (World Trade Center, airlines, tourism) ↓ → slight decrease in GDP CO → “War on Terror”, 2003 US/Iraq war	(CO, BI); (BI, ES); (CO, CO)

^1^ Catastrophe date, location and country ISO code; ^2^ one-to-one interactions (→) used for encoding while other multi-variate consequences (⇒) only mentioned—Symbols ↑ and ↓ mean rise and fall/shutdown, respectively; ^3^ subject to interpretation, here kept as generic as possible.

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
