# Peer review of "Exploring the Space of Possibilities in Cascading Disasters with Catastrophe Dynamics"

_ijerph, 2020, doi:10.3390/ijerph17197317_

Round 1

Reviewer 1 Report

The paper deals with an interesting and timely issue. The use of the Markov chain is plausible, yet one may think of additional technical tools to support the analysis.

The idea of coding historical data is interesting yet it is open to interpretations and thus the scientific merit and soundness may be questionable. I suggest explaining this aspect more and mainly refer to it in discussing the limitations if the study.

The authors should add a section dealing with the limitations of the study. 

Author Response

The Markov chain is a powerful tool and the use here of the fundamental matrix and concept of absorbing state is original in the context of catastrophe dynamics. We plan to extend the study to nonlinear continuous time Markov process (or other more general stochastic processes) models in future studies.

We added a new section, sec. 4 'Limitations and future directions', which includes some of the limitations already addressed in the old Discussion section. The subsection 4.1 'Incompleteness issues in historical data' (lines 270-291) is new and should answer to the review's concerns regarding encoding from historical data. In particular, we discuss the possible biases in the encoding, other biases related to p_ij and the need to build a large database to move from a pilot semi-quantitative analysis to a fully quantitative multi-risk assessment.

Reviewer 2 Report

The authors present an interesting study of cascading failure. However, the reviewer has several concerns about the paper:

  1. What is the development logic of the 24 different topologies in the matrix A (Line 156)? What does each topology represent? It’s better to explain each scenario used in the analysis.
  2. Line 159, what is the unit of timestep? Day? Hour? Minute? Or Seconds? The triggering effect of different events varies in time. For example, an earthquake can trigger a near-field tsunami within 30min, but an earthquake can trigger landslide or building collapse within a couple of minutes. In addition, if the time steps of different triggering events are different, how to justify they can all be represented with a 10*10 matrix A? Without specifying the unit and considering the triggering time difference, the so-called semi-quantitative analysis may generate wrong information.
  3. Figure 3, What does the color code mean? What’s the difference between dark red and orange? What is each color’s range?
  4. Line 21. What do the high in-degree and betweenness centrality mean in this case? The network failure and business interruption can be the result of any type of natural hazards. Why do we need a graph theory analysis to derive knowledge that can be easily derived? The use of the method is not well justified, and the in-depth analysis is required.
  5. The triggering probability of different cascading events is different. For example, the probability of earthquakes lead to landside is different from the landslide to valley fever outbreak (Table 2). Considering them the same would lead to overestimation of the cascading failure risks. This also links to the interaction in Figure 4. It is currently treated as an unweighted graph, but it should be weighted in reality. The conclusion derived from the network analysis can be different when the weights of different links are considered. In addition, probability Pij was used many times. However, no clear quantitative definition of the probability is presented in the paper.

Minor suggestions:

  1. When the author refers to the cases (e.g., Line 167), it’s better to number them in the table as well so it’s easier to track which events they are talking about.
  2. Please add the color legend in figure 3. The figure should be self-contained. It’s not convenient for readers to go back to the previous figure to understand what does this figure mean.

Author Response

We here address the reviewer's concerns point by point:

(1) We describe in more details the development logic lines 172-181: we now explain that the complexity of the patterns increases plot after plot in Figure 2, starting from simple 1-to-1 interactions to 1-to-n, n-to-1 and finally n-to-n interactions. We also mention that no automated rule has been used to create this figure and that its primary role is to illustrate how M derives from A geometrically. To the best of our knowledge, this has never been illustrated, although such visual guide is helpful to better understand Eq. 2.

(2) This is indeed a limitation of our analysis. We added a subsection 'The concept of unit time in the current framework' (lines 327-338) in the new section called 'Limitations and future directions' and explain that while the physical time is lost (simplified into consecutive trials in the proposed discrete Markovian framework), the causal relationship is preserved. We also mention how to incorporate different time steps in future studies. The aim of our work is not to quantify chains of events for multi-risk assessment but to explore, rather qualitatively, the space of possibilities, which is an important prerequisite to more comprehensive dynamical frameworks.

(3) Figure 3 uses the same color range as in Figure 2. Only relative warmness information is useful in such semiquantitative approach where the same probability p is fixed. What needs to be distinguished are patterns of expansion of the space of possibility (in blue) versus patterns of amplification (in warm colors). However, we now add that the scheme for warm colors is proportional to p. Regarding the value of p, see our answer to question #5. The color scheme is now also added to Figure 3 as per request #7.

(4) The centrality measures are not the central result of our study but only a side note corroborating the results obtained from the interaction matrix M. It is quite common to add such measures for the study of adjacency matrices, hence their mention. It was not costly to add those metrics as colors on the nodes of Figure 4. In the text, only one short paragraph is used to describe those metrics to the general reader (lines 97-104). Figure 4 is worthwhile having published since a graph is easier to grasp than, and complementary to, an adjacency matrix.

(5) This is true and the reason why we describe this study as semi-quantitative. Defining realistic conditional probabilities would require a very large database which does not yet exist for such large variety of perils. We explain this constraint in the new subsection 'Incompleteness issues in historical data' (lines 270-291), part of the section on limitations and future directions. It is also true that the case mentioned by the reviewer (outbreak triggered by a landslide) is a rarity. It shows the reader the wide range of possibilities, but also does not impact the main semi-quantitative results of our conclusions which relate to business interruption, network failure and earthquakes. Most other historical cases of Table 1 are relatively common in comparison. This is now clarified in the same subsection (lines 276-282). We re-emphasize the fact that this work is not aimed at direct use in multi-risk. It however provides the framework that could be applied in the future on a larger database from which specific p_ij could be inferred (note that we now express the ad-hoc probability value used in the study, p = 0.1, lines 217 and 239. Other values had been tested with similar patterns always observed). We note that the final result would change but the general patterns would likely be preserved. To make it clear from the start, we added "subject to data incompleteness" in the abstract line 20 (we cannot add more in the abstract due to length constraints). Finally having no weight allows us to focus on general trends and not on the impact of specific high-weight interactions. Again, we were here interested in first-order observations (see the last paragraph of the introduction).

(6) Cases match the topologies of Figure 2. Please notice that the number is already given in parentheses on top of each plot for rapid tracking

(7) The figure has been modified accordingly.

Reviewer 3 Report

Overview:

In this manuscript, the authors investigated the space of possible interactions between 19 types of loss-generating events, presented the impact of 24 topologies of one-to-one interactions, and computed the matching matrix by using 29 historical cases of cascading disasters. This manuscript is interesting but requires a major, good revision. Some detailed comments are listed below.

Detailed Comments:

  1. Overall, the model representation could be improved; for example, it would be great if the authors mentioned that q = 1 - p at line 72, the dimensions of matrices P, Q, and A, the definition of n_perils at line 74 (though it should be the number of perils), and the number of zeros at line 77 (though it should be n_perils).
  2. The standard style should be used for figures; compare figures 1, 2, and 4.
  3. What is the meaning of "With G finite" at line 111?
  4. Recheck the reference style; for example, "Mignan et al. described in [4]" at line 120.
  5. What is the difference between perils and catastrophes? If there is no specific difference, it would recommend to choose either one of those two and to use it in the manuscript.
  6. It is not easy to understand Table 2.
  7. Recheck some typos; for example, "section 3.1" at line 151 and "(Eq. 2)" at line 164.
  8. The procedures to draw the Figure 2 and Figure 3 are not presented. How to use the equation (2) for Figure 2? How to use Table 2 for Figure 3? There is no way to check if the figures are correct.  It would recommend to provide any example(s) showing the matrices A, A bar, and M.
  9. It is difficult to figure out Figure 4, especially the color representations for links.
  10. The use of absorbing Markov chains is highlighted in this paper, but rather most of cases use the matrix A bar, an adjacency matrix without absorbing states. What is the principal contribution of the absorbing Markov chains in this paper? It would be great to provide the practical meaning of transient states and absorbing states in the matrix.

Author Response

We here address the reviewer's concerns point by point:

(1) Please note that q is not 1-p, q being a rate and p a probability. We however now relate q to Q line 74. The equation line 69, by definition, means that all matrices have the same dimensions. See answer to question #10 for further clarifications on A and n_perils.

(2) We updated Figure 1 to be as close as possible to the style of Figures 2 and 4. Font size, colour tone, matrix grid colour, edge width have been changed.

(3) Here we refer that the study only considers a finite dimensional model where the state space is discrete and has finite dimensionality. In a more general setting, the fundamental equation may not be a trivial problem to solve.

(4) We removed 'Mignan et al' to only keep [4]

(5) Here a peril is the type of the event, i.e. earthquake, social unrest, etc. while a catastrophe is the combined result of a chain of events. We did reread carefully the text and found no ambiguity or misuse of the two terms.

(6) We used the same format as in ref. [4]. We do not see how to simplify the Table further as the complex interactions are already simplified to the most fundamental events possible. Please have a look at the footnotes which provide some additional information. We added more details about the up-arrow and down-arrow symbols lines 58-59.

(7) We're sorry to say that we could not find any typo where mentioned. Eq. (2) is defined in section 2.1 but it is in section 3.1 that it is first applied. We hope this answers to your question.

(8) All the data and method necessary to reproduce those figures are in the text. Take the matrix A^(1) for example and encode in your programming language of choice a p_ij value, e.g. 0.1 (line 217), on the diagonal. Application of Eq. 2 directly provides M as a function of this matrix A. A repeat of the operation would recreate Fig. 2. As for the topologies chosen in Fig. 2, the logic behind those choices is now explained lines 172-181 - Regarding Fig. 3, the left matrix is Abar based on the data. The matrix A would just include an additional row and column, filled as defined line 79 (same for Mbar now defined, see answer to question #10). Again, application of Eq. (2) on A gives M, which is the matrix on the right in reduced form (i.e. minus the last row/column). Matrix A was simply encoded from the information in the last column of Table 2. This is now clarified lines 224-226.

(9) There is no colour representation for the links, reason why they are all shown in grey. To read the graph, simply choose an event represented by a vertex and follow the directed edges to discover possible interactions. This is an alternative way to the adjacency matrix to visualise interactions.

(10) We agree that it was misleading, and we have now cleaned the notation. We now always use A and M, except when referring to the plots which show the same matrices, but without the last row/column which provides no information except for the abstract outflow event. Those are then the reduced matrices Abar and Mbar. All calculations use A to get M, with A including the absorbing states. It allows the catastrophe, or chain-of-events, to be finite but also conservation of probabilities. Abar is just used to avoid representing the outflow event column in Figs. 2,3 and for further convenience, as this is from Abar that the graph is built in Fig. 4. The term "absorbing state" is only mentioned 6 times throughout the article, so this is not so much of a highlight, but a mathematical necessity. See the following lines with some clarifications on Abar and Mbar: lines 121-123, 169-170, 172, captions of Figs 2-3

Round 2

Reviewer 2 Report

The author has addressed all of my comments. 

Reviewer 3 Report

N/A